# Natural allelic variation of *FRO2* modulates *Arabidopsis* root growth under iron deficiency

Santosh B. Satbhai[1], Claudia Setzer[1], Florentina Freynschlag[1], Radka Slovak[1], Envel Kerdaffrec[1] & Wolfgang Busch[1,2]

Low availability of Fe significantly limits crop yields in many parts of the world. However, it is largely unknown which genes and alleles adjust plant growth in Fe limited environments. Using natural variation of a geographically restricted panel of *Arabidopsis thaliana* accessions, we identify allelic variation at the *FRO2* locus associated with root length under iron deficiency. We show that non-coding sequence variation at the *FRO2* locus leads to variation of *FRO2* transcript levels, as well as ferric chelate reductase activity, and is causal for a portion of the observed root length variation. These *FRO2* allele dependent differences are coupled with altered seedling phenotypes grown on iron-limited soil. Overall, we show that these natural genetic variants of *FRO2* tune its expression. These variants might be useful for improvement of agronomically relevant species under specific environmental conditions, such as in podzols or calcareous soils.

[1] Gregor Mendel Institute (GMI), Austrian Academy of Sciences, Vienna Biocenter (VBC), Dr Bohr-Gasse 3, Vienna 1030, Austria. [2] Salk Institute For Biological Studies, Plant Molecular And Cellular Biology Laboratory, 10010 N Torrey Pines Rd, La Jolla, CA 92037, USA. Correspondence and requests for materials should be addressed to W.B. (email: wbusch@salk.edu).

ron (Fe) is an essential element for both plants and animals. In plants, it is vital for photosynthesis, respiration and a large number of other biological processes and its availability is an important determinant of plant growth. The iron that is taken up by plants is the basis for almost all iron in the food chain. Furthermore, the majority of the global human population relies almost entirely on Fe from plant sources. Therefore, understanding how plants coordinate their growth, as well Fe uptake and storage, with the availability of Fe has important implications for plant growth as well as human nutrition.

Despite its high abundance in the earth's crust, Fe is frequently inaccessible to plants as it tends to form insoluble ferric hydroxide complexes under aerobic and alkaline conditions[1]. Therefore, plants grown where Fe is limited, such as in calcareous soils, often face Fe deficiency, leading to reductions in growth, crop yield and quality[2]. Under low Fe availability, iron deficiency chlorosis (IDC) is a common growth limiting condition for plants[3]. Inversely, excess Fe is also toxic, as Fe leads to DNA damage, production of reactive oxygen species (ROS), and other cellular stresses[4]. Therefore, tight regulation of Fe homeostasis is essential and is strictly controlled by regulating iron uptake, transport and storage.

Plants have evolved two main strategies to solubilize and absorb Fe from the soil: a reduction strategy (Strategy I) and a chelation strategy (Strategy II)[5–7]. Dicot plants, such as Arabidopsis (Arabidopsis thaliana), pea (Pisum sativum), soybean (Glycine max), tomato (Solanum lycopersicum), as well as non-graminaceous monocots employ a reduction-based strategy, where acidification of the rhizosphere occurs by proton extrusion. This acidification helps to increase the solubility of $Fe^{3+}$ complexes in the soil, followed by the reduction of $Fe^{3+}$ to $Fe^{2+}$ and the subsequent uptake of $Fe^{2+}$ in the roots[8]. Strategy II is used by graminaceous monocots, such as rice (Oryza sativa) and maize (Zea mays). These plants use a chelation strategy in which they exude phytosiderophores that chelate $Fe^{3+}$ and the resulting complexes are imported by the roots[9,10].

Key molecular mechanisms for Strategy I were elucidated in Arabidopsis and tomato and several key components along with their role in Fe homeostasis have been described. Many of the genes involved in Fe acquisition strategies are transcriptionally upregulated in response to Fe deficiency. Under Fe deficiency, protons are released into the rhizosphere by the proton-translocating ATPase AHA2 (arabidopsis plasma membrane $H^+$-ATPase isoform 2), which results in local rizhosphere acidification[11,12]. After acidification, $Fe^{3+}$ is reduced to $Fe^{2+}$ by the membrane-bound ferric chelate reductase FRO2 (ferric reductase oxidase 2). $Fe^{2+}$ is then taken up into root epidermal cell layers by the specialized root $Fe^{2+}$ iron transporter IRT1 (refs 13,14). The reduction from $Fe^{3+}$ to $Fe^{2+}$ appears to be the crucial, rate-limiting step in Fe uptake in Arabidopsis[15]. Apart from these molecular processes, root growth and development respond quickly to altered iron concentrations. Fe deficiency triggers the formation of ectopic root hairs, which play an important role in Fe uptake by increasing the absorptive surface area of the root[16]. It has been also shown that lateral root architecture and length are altered in response to localized Fe supply in Arabidopsis[17]. In particular, both the length of the primary root and the number of lateral roots increase under moderate Fe deficiency; however, they significantly decrease under severe Fe deficiency[18]. Transcriptome profiling of roots of different Arabidopsis accessions showed that there are genotype dependent differences in the early responses and adaptation to iron deprivation[19]. Taken together, these studies suggest that the molecular and physiological response to varying levels of iron is genetically regulated. However, almost nothing is known about the genetic basis for this. For instance, it is not known to which extent natural genetic variation modulates root growth in response to Fe deficiency, which genes and variants are responsible for this, or how these genes might be involved in local adaptation to environments with constraints on Fe availability.

Here, using natural variation of early root development under Fe deficiency (−Fe) conditions among natural Arabidopsis accessions from Sweden, we show remarkable variation for this trait. Using genome-wide association (GWA) mapping we identify allelic variation at the FRO2 locus as a key determinant of this response. Furthermore, we find that non-coding polymorphisms in the FRO2 alleles lead to variation of FRO2 transcript levels, as well as ferric chelate reductase activity, and are causal for the observed variation.

## Results

**Natural variation of Arabidopsis root length under -Fe.** We set out to identify genetic components that regulate plant growth upon Fe deficiency. We reasoned that a local set of Arabidopsis accessions from highly contrasting environments might increase our ability to identify causal genetic variants that are important for local adaptation, including those related to Fe acquisition. An Arabidopsis population from Sweden contains two distinct groups that have adapted to very different environments[20,21]. The environmental differences between locations from which the accessions were collected include soil classes with different characteristics in retaining micronutrients (Supplementary Fig. 1). Therefore, we assessed root growth of 134 Swedish Arabidopsis thaliana accessions under Fe deficiency (Supplementary Data 1). We observed extensive natural variation of root length under Fe deficient conditions (Fig. 1a, Supplementary Fig. 2). The major proportion of the observed root length variation under −Fe cannot be explained by developmental differences between accessions, as we did not find a strong correlation between root length under +Fe and −Fe conditions (Supplementary Fig. 3). Moreover, subsequent genome-wide association studies (GWAS) for root length traits revealed no overlapping significantly associated loci between +Fe and −Fe conditions (Supplementary Fig. 4). Overall, these results demonstrate that even a limited, local sample of Arabidopsis accessions displays extensive and specific natural variation in response to Fe deficient growth conditions.

**GWAS links root length variation under -Fe to FRO2.** To identify genomic regions underlying the growth response to −Fe, we conducted GWASs on root lengths of the accessions for each day of a 5-day time course. We identified multiple significantly associated SNPs using a 5% FDR threshold calculated by the Benjamini-Hochberg-Yekutieli procedure (Supplementary Fig. 4). The most significant of these associations was located on chromosome 1 for the median total root length at day 1 and day 2 (Fig. 1b; Supplementary Fig. 4). The most significantly associated SNP (marker SNP) of this peak on chromosome 1 was located at position 207,099 and its reference (Col-0) allele was associated with short roots under −Fe, while its non-reference variant was associated with long roots. To identify the causal gene underlying this association, we first measured the expression of the three genes within an approx. 5 kb distance surrounding the GWA peak under iron sufficient conditions. We included accessions that displayed either short roots (termed Group 1: Boo2-3, T1070, and Grön 14) or long roots in −Fe medium (termed Group 2: TNY 04, TDr-16, and TV-10). Additionally we included an accession with an intermediate root length (T980). We found that, of the three genes surrounding the peak, only AT1G01580 showed a

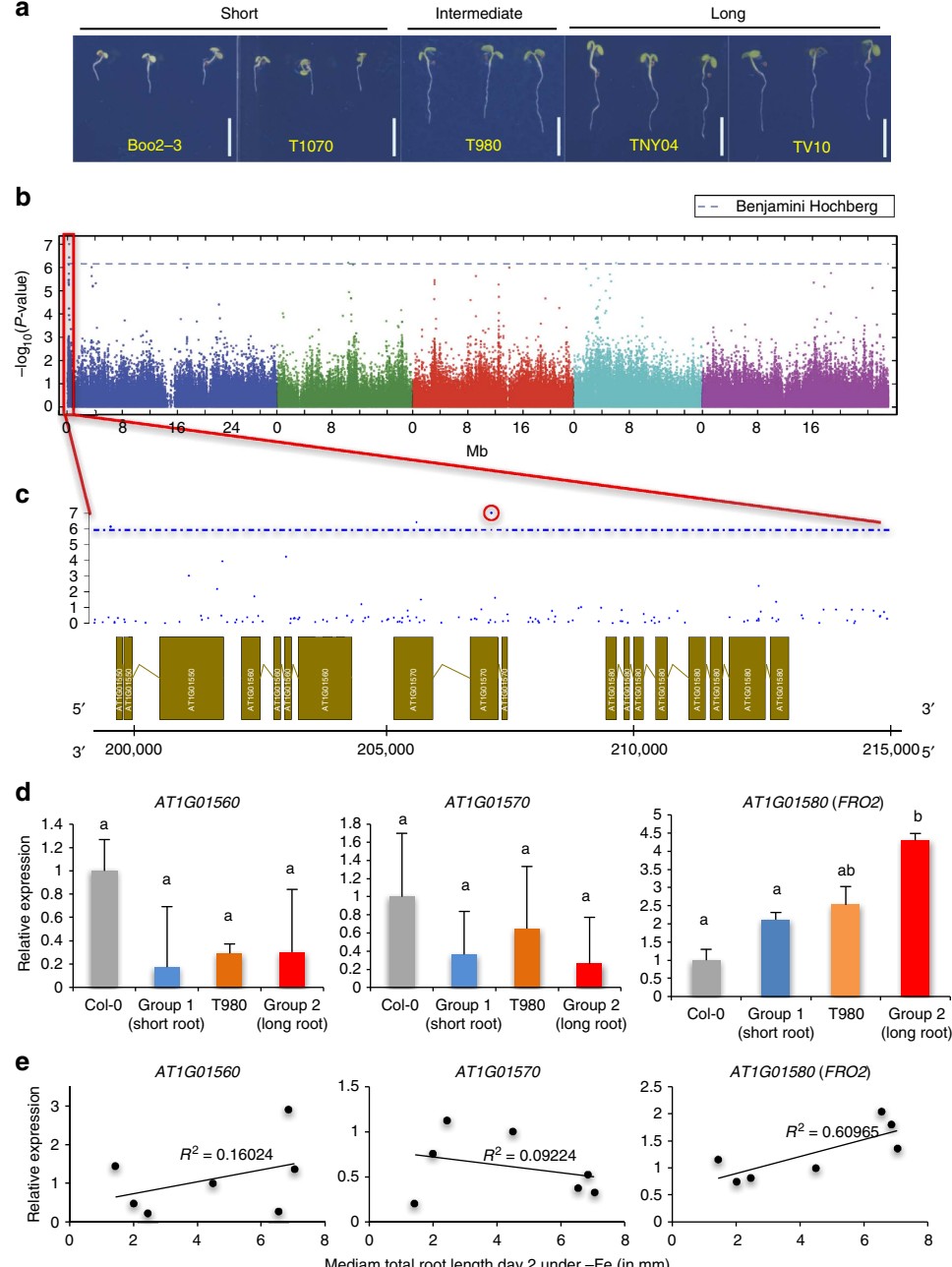

**Figure 1 | GWAS of root growth under Fe-deficient conditions.** (**a**) Representative accessions of *A. thaliana* that show short, intermediate, and long root phenotypes on Fe deficient medium 6 DAG. Scale bars, 1 mm. (**b**) Manhattan plot for the SNP associations to median root length on day 2. Analysis performed on GWAPP[42]. Chromosomes are depicted in different colours. The horizontal blue dash-dot line corresponds to a nominal 0.05 significance threshold after Benjamini Hochberg (False Discovery Rate) correction. Red box indicates the region containing the significantly associated locus. (**c**) The genomic region surrounding the significant GWA peak. Top, $-\log_{10}$ P-values of association of the SNPs. Bottom, gene models in genomic regions. The x-axis represents the position on chromosome 1. (**d**) Gene expression of AT1G01560, AT1G01570, and *FRO2* in accessions displaying extreme root lengths (Group 1: Boo2–3, T1070, and Grön 14, short; T980, intermediate; and Group 2: TNY 04, TDr-16 and TV-10, long) as measured by qRT-PCR. Expression levels were normalized to expression in Col-0. Error bars: s.e.m. For qRT-PCR analyses, seeds were evenly placed on the mesh in a single row at a density of ~20 seeds per cm in two rows. Whole roots were sliced off and collected in liquid nitrogen. Around 50 plants were pooled together; data from three independent biological replicates each with two technical replicates are expressed as s.e.m. The letters a and b indicate significant differences between mRNA expression levels (determined by one-way ANOVA and Tukey HSD ($P < 0.05$, $n = 3$) (**e**) Scatter plots of gene expression (y-axis) and total root length (x-axis) in accessions displaying extreme short (Boo2-3, T1070, and Grön 14), intermediate (T980) and long root lengths (TNY 04, TV-10, and TDr-16). The lines represent the result of linear regression. $R^2$: coefficient of determination of the linear regression.

significant differential expression between group 1 and group 2 ($P < 0.007$, one-way ANOVA, Fig. 1d). Moreover, the transcript level of this gene was highly correlated with root length under $-$ Fe conditions ($R^2 = 0.60$, coefficient of determination of the linear regression), while transcript levels of the other two genes were not or only marginally correlated (Fig. 1e). This unbiased analysis clearly indicated that AT1G01580 is the best candidate gene for being causal for the natural variation of root

length in response to $-$Fe. Importantly, its known function is highly consistent with this model, as AT1G01580 encodes *FRO2*, a major regulator of iron homeostasis. *FRO2* is the low-iron-inducible ferric chelate reductase responsible for reduction of iron at the root surface, and it has been shown that *frd1-1* mutants (*FRO2* loss-of-function) display impaired plant growth under Fe deficiency as compared with wild type[22]. While our analysis clearly pointed towards *FRO2* as the best candidate gene, another gene (*FRO1*—AT1G01590) encoding a ferric-chelate reductase is located in tandem with *FRO2*. To test whether *FRO1* expression was correlated with root growth responses in the accessions, we measured its transcript levels in extreme accessions under $+$Fe (Supplementary Fig. 5a) and $-$Fe (Supplementary Fig. 5b) conditions. Consistent with previous reports[23], we did not find any significant expression change of *FRO1* under $-$Fe condition in Col-0 or accessions of group 1 or group 2 (Supplementary Fig. 5). We also didn't find a *FRO1* expression pattern consistent with the root phenotypes of group1 and group 2 accessions (Supplementary Fig. 5). We therefore concluded that an involvement of *FRO1* in the observed natural variation is very unlikely.

The function of our top candidate gene, *FRO2*, has been tied to its ferric chelate reductase activity. Induction of this activity is critical for iron uptake under $-$Fe condition[15,22]. To test whether root length variation is not only associated with an increased *FRO2* transcript level but also with increased ferric chelate reductase activity, we assayed the ferric chelate reductase activity in accessions with short roots (Group 1) and long roots (Group 2) under iron-sufficient and iron-deficient conditions. We found a significantly higher induction of ferric chelate reductase activity under $-$Fe in Group 2 accessions (Fig. 2a). Furthermore, we found that ferric chelate reductase activity in contrasting accessions was highly correlated with root length ($R^2 = 0.664$, coefficient of determination of the linear regression) under $-$Fe conditions (Fig. 2b), as well as with *FRO2* transcript levels (Fig. 2c; $R^2 = 0.662$, coefficient of determination of the linear regression).

**FRO2 regulates root length under Fe deficiency.** Our analyses strongly suggested that the *FRO2* gene is the causal gene for the root growth response to Fe deficiency. Furthermore, the data also suggested that its expression level is involved in determining root length under Fe deficiency. However, the top GWAS SNP (Fig. 1c) was located in AT1G01570 and not in the *FRO2* gene. While it has previously been shown that *FRO2* function is required for plant growth and development under Fe deficiency conditions[22] and overexpression of the *FRO2* gene enhances the tolerance to Fe deficiency[15], a role for *FRO2* in root length control had not yet been reported. To test whether *FRO2* or AT1G01570, or both, are involved in root length determination upon $-$Fe, and thereby gain additional evidence as to which gene is causal, we quantitatively tested the root growth of loss of function lines for both genes on full and Fe-free medium. Unlike the screening conditions, in which we prepared the medium using iron-free nutritive solution, regular agar, and the chelator FerroZine to chelate residual iron from the agar, we performed these experiments with metal-depleted agar containing an Fe-free nutritive solution and without chelating agent (Fe-free). This allowed us to exclude confounding due to FerroZine and potential traces of iron. In these experiments, we saw only a slight root length reduction of *fro2* under control conditions as compared with WT (average root length reduced by 10.7%), while *fro2* mutant plants showed a very strong reduction of root length when grown in Fe-free conditions as compared with WT plants (average root length reduced by 58.3%, Fig. 3a,b). This was not

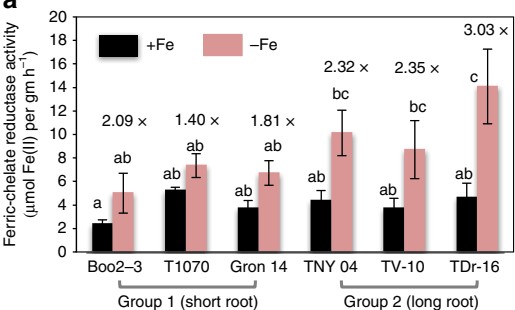

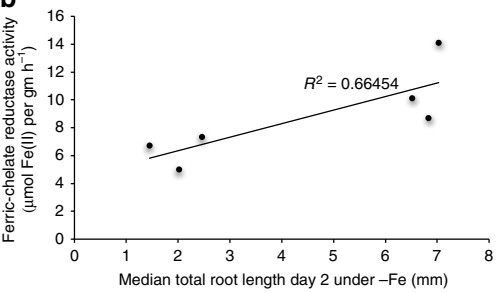

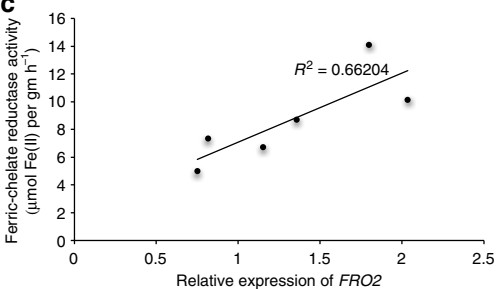

**Figure 2 | Assays of ferric chelate reductase activity in extreme accessions.** (**a**) Assays of ferric chelate reductase activity in accessions with short roots (Group 1: Boo2-3, T1070, and Grön 14) and accessions with long roots (Group 2: TNY 04, TV-10, and TDr-16). Plants were grown on standard medium for 14 days and then transferred to either iron-deficient (red bars) or -sufficient (blue bars) medium for 3 days before the assay. Around 10 plants were pooled together; data from three independent biological replicates each with two technical replicates are expressed as s.e.m. The letters a and b indicate significant differences (determined by one-way ANOVA and Tukey HSD ($P < 0.05$, $n = 3$). (**b**) Scatter plots of ferric chelate reductase activity (*y*-axis) and total root length (*x*-axis) and (**c**) *FRO2* transcript levels (*x*-axis) in accessions displaying extreme short (Boo2-3, T1070, and Grön 14) and long root lengths (TNY 04, TV-10, and TDr-16). The lines represent the result of linear regression. $R^2$: coefficient of determination of the linear regression.

observed in the *at1g01570* mutant, which showed a pronounced shorter root phenotype than WT in both full and Fe-free conditions (Fig. 3a,b). Consistently, the *fro2* mutant line displayed retarded growth when grown on alkaline soil (pH 7.0–8.0), a growth condition which leads to Fe deficiency due to reduced Fe solubility at the higher pH (Fig. 3c). This growth inhibition was not observed in *at1g01570* knockout plants (Fig. 3c). Consistent with the role of *FRO2* in determining root length under $-$Fe, its overexpression led to a significant increase in root length under $-$Fe compared with wild type. This increase was specific for $-$Fe as we did not observed significant differences under $+$Fe conditions (Supplementary Fig. 6a,b). In sum, our data clearly show that *FRO2* expression regulates root growth in $-$Fe conditions.

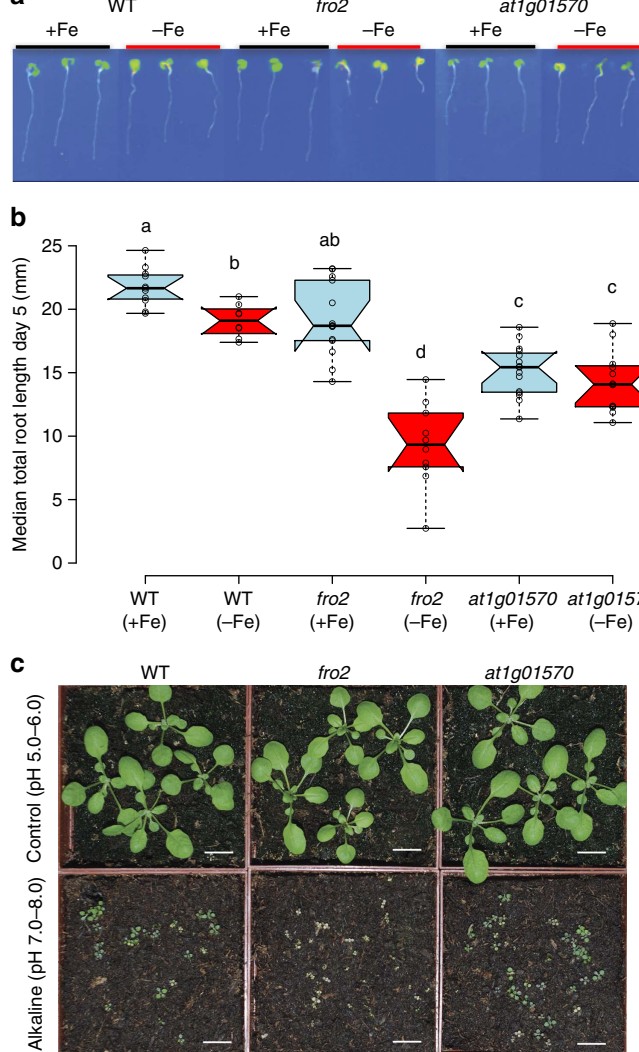

**Figure 3 | FRO2 regulates root growth under Fe deficiency.**
(**a**) Representative seedlings of wild-type (Columbia Col-0), fro2 mutants, and at1g01570 at 6 DAG grown on Fe sufficient (1× MS) or Fe-free medium. Scale bars, 1 mm. (**b**) Boxplot of total root length of WT, fro2 and at1g01570. Horizontal lines show the medians; box limits indicate the 25th and 75th percentiles; whiskers extend to 5th and 95th percentiles. Letters a, b, c and d indicate significantly different values at $P < 0.05$ determined by one-way ANOVA and Tukey HSD. (**c**) Plant vigor of WT, fro2, and at1g01570 on normal soil pH (5.0–6.0) and alkaline pH (7.0–8.0; low Fe availability) Plants were grown on normal or alkaline soil for two weeks. White scale bars, 10 mm.

**Allelic variation of FRO2 determines root growth under −Fe.**
While our data strongly suggested that different alleles of FRO2 are causal for the observed natural variation of root growth responses to iron deficient conditions, they were still based on correlation. To test whether allelic variation of the FRO2 locus is causal, we complemented the fro2 mutant line with different alleles from contrasting accessions. In such transgenic approaches, transgene insertion sites can significantly influence expression levels. We therefore wanted to test a large number of independent insertions to account for such random effects. We generated independent transgenic fro2 mutant background lines that were transformed with the empty vector (control) or pFRO2:FRO2 alleles from either Boo2-3 (accession with short root length in −Fe), T980 (intermediate root), or TNY 04

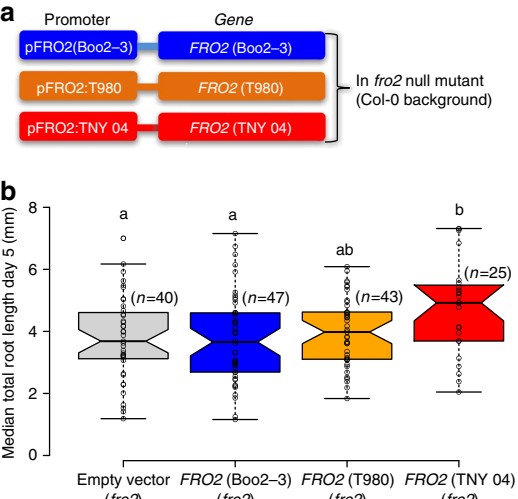

**Figure 4 | Natural allelic variation of FRO2 locus causes phenotypic variation of root length in Fe deficiency conditions.** (**a**) Schematic of transgenic constructs used for complementation of the fro2 null mutation (Col-0 background). (**b**) Box plot shows median total root length of 6 DAG old seedlings grown in Fe deficient conditions. Horizontal lines show the medians; box limits indicate the 25th and 75th percentiles; whiskers extend to 5th and 95th percentiles. Letters a and b indicate significantly different values at $P < 0.05$ determined by one-way ANOVA and Tukey HSD.

(long root) (Fig. 4a). To phenotype a large number of transgenic lines at the T1 generation without the need for antibiotic selection, we included a Pro35S:PM-mCherry reporter gene in the construct as a visual marker so that we could sort seeds containing the transgene based on fluorescence. T1 lines carrying a FRO2 allele from an accession with long roots (TNY 04) in the fro2 background developed significantly longer roots ($n = 25$, $P < 0.02$, one-way ANOVA) as compared with those carrying empty vector ($n = 40$), as well as fro2 plants complemented with a FRO2 allele from an accession with short roots (Boo2-3, $n = 47$) (Fig. 4b). The T1 lines that were complemented with a FRO2 allele from the accession with an intermediate root length (T980, $n = 43$) displayed an intermediate phenotype (Fig. 4b). These data clearly demonstrate that FRO2 allelic variation significantly contributes to natural variation of root growth upon − Fe.

**Non-coding polymorphisms at the FRO2 locus are causal.** After we had shown that allelic variation of FRO2 is causal for the natural variation in root growth responses under − Fe, we wanted to identify the most likely causal sequence polymorphisms. We therefore compared the pattern of polymorphisms of the 6 accessions with the longest roots under − Fe and that possessed the non-reference allele of the marker SNP with the 6 accessions with the shortest roots having the reference allele at the marker SNP. Interestingly, apart from the marker SNP, there were no common SNPs within these groups. However, we found that the accessions with the longest roots and the non-reference marker SNP seemed to represent two different groups with respect to the polymorphisms in the FRO2 region (Supplementary Fig. 8). This suggested the presence of allelic heterogeneity (multiple FRO2 alleles that cause the same effect). We therefore conducted a haplotype analysis on a 50-kb region of chromosome 1 centered on FRO2 (Supplementary Fig. 9). We found four main haplotypes in the accessions that we used in this study (Supplementary Fig. 9). Accessions containing two of the main haplotypes (LR1 depicted in dark green and LR2, depicted in light pink) tend to have longer roots than the other

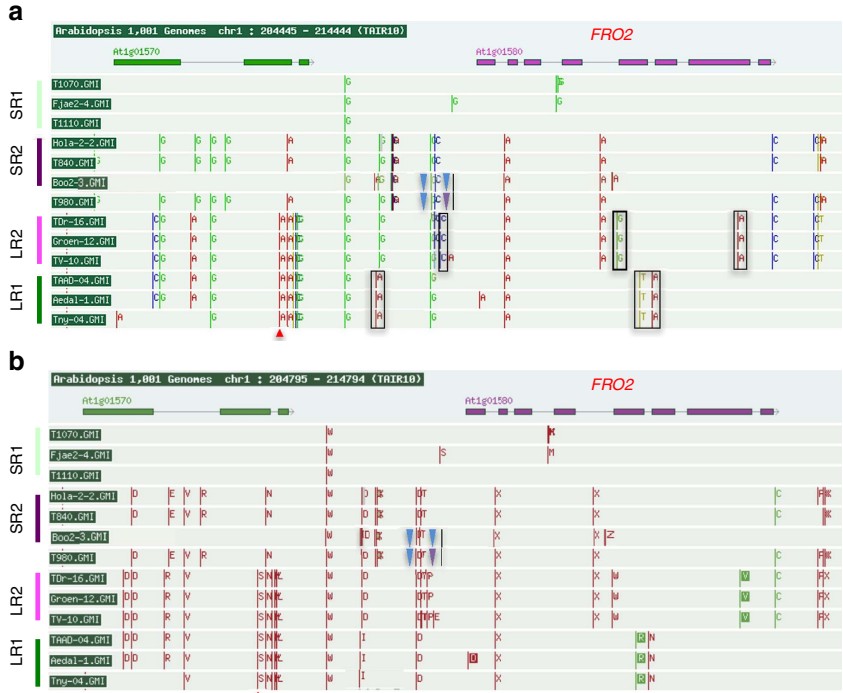

**Figure 5 | SNP polymorphisms around the *FRO2* locus in extreme accessions of distinct *FRO2* haplogroups.** SNP polymorphisms (**a**) and amino acid changes (**b**) surrounding the *FRO2* locus in three representative accessions from the four main haplotypes (faint green (SR1) and dark pink (SR2) haplotypes with short root phenotype and faint pink (LR2) and dark green (LR1) with long root phenotype). Red arrow: location of the significant GWA SNP. Black boxes: Most likely candidate SNPs underlying the allelic effects. SNP polymorphism in *FRO2* regulatory and coding region is constructed based on Sanger sequencing data. SNP polymorphisms in Boo2-3, T980, and TNY 04 accessions are modified based on Sanger sequencing data (Black line: 1 bp deletion, blue triangle: 2 bp insertion, purple triangle: 3 bp insertion). Only genomes that were available on the SALK 1,001 genomes browser (http://signal.salk.edu/atg1001/3.0/gebrowser.php) as of September 2016 were considered.

haplotypes. In the light of these multiple haplotypes, we re-analysed the SNP polymorphism pattern (Fig. 5) and found that the two haplotypes associated with longer roots upon −Fe share distinct polymorphisms. The LR2 haplotype exhibits three SNPs: in the *FRO2* promoter region, the 4th intron, and the 7th exon (however, this SNP does not change the amino-acid sequence, Fig. 5a,b). The LR1 haplotype shows three SNPs: in the promoter region, the 5th exon (also not leading to an amino acid substitution), and in the 5th intron (Fig. 5a,b). Overall, these data show that two different alleles of *FRO2* are associated with increased root growth on −Fe but that these two alleles could be detected using the same marker SNP.

The polymorphism pattern suggested that non-coding variation was responsible for the observed *FRO2* allele dependent natural variation. We therefore wanted to test whether allelic variation of the *FRO2* locus caused altered *FRO2* transcript levels as well as ferric chelate reductase activity. As the data from the genome re-sequencing can contain errors, we first reconfirmed the 1,001-genomes SNPs and the absence of insertions and deletions in the coding sequence of the *FRO2* alleles that we had used for complementation using Sanger sequencing (Supplementary Fig. 10). To have enough biological material for these assays, we generated two independent T3 homozygous lines from T1 lines (as shown in Fig. 4) of *fro2* carrying the empty vector or complemented with the *FRO2* allele from the Boo2-3 (SR2 haplotype) or TNY 04 (LR1 haplotype) accessions. We first tested whether these T3 lines were representative of the phenotypes observed in the T1 lines. While all of the T3 lines were more vigorous than their T1 counterparts, the genotype dependent differences between the lines were indistinguishable from those observed in the T1 generation (Fig. 4b); single

insertion lines of *fro2* lines complemented with the TNY 04 (long root) *FRO2* allele had significantly longer roots under −Fe than *fro2* lines carrying the empty vector or the Boo2-3 (short root) *FRO2*-allele (Fig. 6a,b). Concomitantly, *FRO2* expression levels and ferric chelate reductase activity were also significantly higher in the lines carrying the TNY 04 *FRO2* allele compared with empty vector control and lines carrying the Boo2-3 allele (Fig. 6c,d). Finally, a similar pattern emerged when these lines were grown on soil at an alkaline pH (7.0–8.0) (Fig. 6e). Taken together, these data clearly show that natural allelic variation at the *FRO2* locus causes differential expression of *FRO2* mRNA, ferric chelate reductase activity, and root length under −Fe conditions. Moreover, they also show that these *FRO2* allele dependent differences are coupled with altered seedling phenotypes on iron-limited soil.

## Discussion

In this study, we report significant natural variation of root length in a geographically limited panel of *Arabidopsis* accessions under −Fe (Supplementary Fig. 2). We identify allelic variation in the *FRO2* gene to be causal for this variation (Fig. 4). This allelic variation is non-coding and leads to changes in expression of *FRO2* mRNA and altered ferric reductase activity.

As many GWAS associations relate to uncharacterized genes, it is intriguing that we uncovered a major player in the Fe homeostasis pathway to be responsible for the natural variation of root length we observed in −Fe conditions. The *FRO2* gene is a necessary component for the Fe uptake pathway and encodes a membrane bound enzyme that reduces Fe to the bioavailable form[1,22]. The *frd1* mutants (null mutants of *FRO2* gene) were

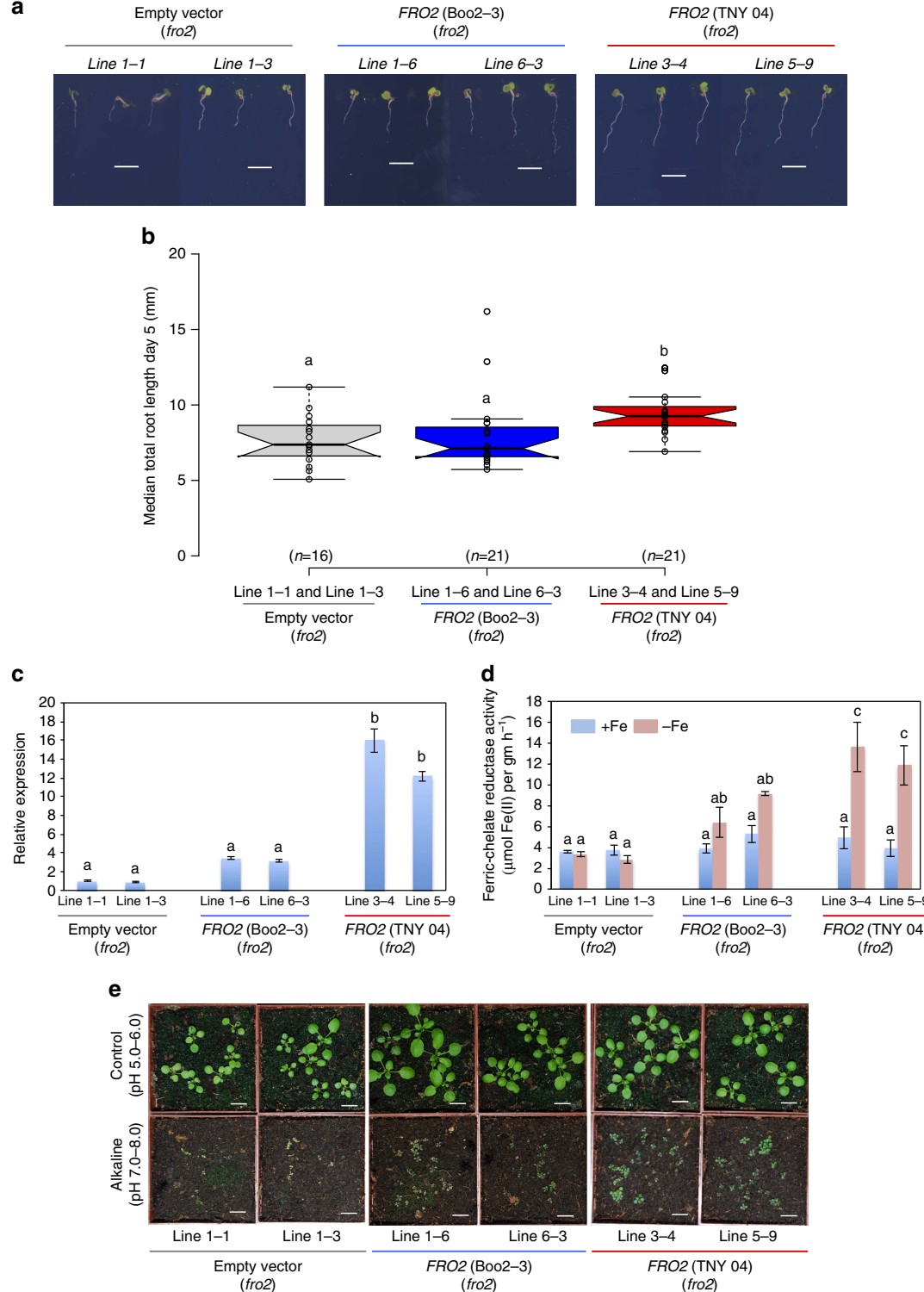

**Figure 6 | *FRO2* regulates root length in transgenic lines under −Fe.** (**a**) Representative seedlings of two independent T3 homozygous lines carrying empty vector (line 1–1 and line 3–3), Boo2-3 alleles (line 1–6 and line 6–3) and TNY 04 alleles (line 3–4 and line 5–9) at 6 DAG grown on Fe-deficient medium (+300 μM Ferrozine) Scale bars: 10 mm. (**b**) Boxplot of total root length of empty vector lines, lines carrying Boo2-3 and TNY 04 allele. Horizontal lines show the medians; box limits indicate the 25th and 75th percentiles; whiskers extend to 5th and 95th percentiles. Letters a and b indicate significantly different values at $P < 0.05$ determined by one-way ANOVA and Tukey HSD. (**c**) *FRO2* expression in transgenic lines as measured by qRT-PCR. Expression levels were normalized to expression in empty vector control lines. qRT-PCR analyses were done as mentioned above, data from four independent biological replicates each with two technical replicates are expressed as s.e.m. The letters a and b indicate significant differences between mRNA expression levels (determined by one-way ANOVA and Tukey HSD ($P < 0.05$, $n = 4$). (**d**) Assays of ferric chelate reductase activity in transgenic lines. Assays were performed as mentioned above. The letters a, b and c indicate significant differences (determined by one-way ANOVA and Tukey HSD ($P < 0.05$, $n = 3$). (**e**) Plant vigor of transgenic lines on normal pH soil (5.0–6.0) and alkaline pH soil (7.0–8.0; low Fe availability). Plants were grown on normal or alkaline soil for two weeks. White scale bars: 10 mm.

previously shown to display major plant growth defects when grown under − Fe condition[22]. We expanded this phenotype by showing that fro2 mutant plants developed significantly shorter roots only when grown under − Fe conditions (Figs 3 and 4). Conversely, roots of WT plants are only slightly shorter on − Fe than under + Fe conditions (Fig. 3). This is quite remarkable, as the medium had been completely depleted of iron and therefore no usable iron should be present. As there is no iron for WT or fro2 mutants to take up, we would expect similar growth effects in both lines if the phenotype was related to iron availability. As the phenotypes were different, but the absence of Fe alone does not dramatically restrict root growth of WT at this stage of root development, this therefore suggests that either FRO2 has a regulatory function in addition to its biochemical function or that the Fe-loading of seeds in the maternal plants is responsible for this. In the latter case more substantial Fe reserves in the seeds would allow seedlings to sustain growth longer on − Fe medium. Both models are consistent with the observation that plants overexpressing FRO2 grew at a much higher rate than WT during two weeks on Fe deficient medium[15]. However, as mutants of other iron acquisition and homeostasis related genes also show root growth effects on − Fe (refs 24–26), an effect of FRO2 activity on the Fe-loading into the seed seems more likely.

Using a large number of independent T1 lines for our allelic complementation experiment (Fig. 4), we could clearly show that the natural variation of root growth under Fe deficiency in the set of Swedish accessions is, to a significant extent, determined by allelic variation of the FRO2 locus. The correlation of the steady state expression level of FRO2 in contrasting accessions, the polymorphism pattern, and the altered mRNA level in the complementation lines all show that FRO2 expression levels underlie this phenotypic variation. Potential regulatory SNPs underlying this variation are located in the promoter and intronic and exonic regions of FRO2 (Fig. 5a; Supplementary Fig. 10). Consistent with the hypothesis that intronic and exonic SNPs could affect the steady state expression of FRO2, it has been shown that FRO2 mRNA levels are post-transcriptionally regulated, as FRO2 expressed under the constitutive 35S promoter is still regulated by Fe deficiency[15]. Therefore, the exonic and intronic SNPs we identified are prime candidates for this post-transcriptional regulation.

The relationship between natural genetic variation, consequential phenotypic variation, and the local environments from which accessions were isolated can open an avenue to understanding adaptation to environmental variables. We observed a significant correlation between root lengths of Swedish accessions under Fe-deficient conditions and temperature variables of the environments where these accessions naturally grow (Supplementary Fig. 11a, Supplementary Table 1). Most of the accessions with long roots upon − Fe come from a region of lower temperature and higher latitude, while most of the short root accessions come from the milder South of Sweden (Supplementary Data 2). However, this correlation is lost when using a mixed model to correct for population structure, an issue we expected as the clinal variation of the phenotype (mainly short in the south and long in the north) and the population structure follow a similar pattern (Supplementary Table 2). When performing a latent factor mixed model (LFMM) associating SNPs and climate variables while accounting for population structure[27] on the RegMap panel[28] we found a significant association ($P < 0.05$) between the top GWAS marker SNP and the mean annual temperature, however, after multiple testing correction, it was not significant at a genome wide scale. While these analyses are therefore not conclusive, there are additional reasons to believe that FRO2 allelic variation is related to temperature. For instance, it was shown that FRO2 function is required in Glycine-Betaine (GB) mediated chilling

tolerance in Arabidopsis[29] and Fe deficiency chlorosis has been associated with soil temperature[30,31]. Another potentially adaptive role consistent with the data is hinted at by the differences in iron retention of the soil classes occurring in northern and southern Sweden. The vast majority of accessions containing a FRO2 allele associated with fast growth under − Fe were collected from areas covered by Podzol (Supplementary Fig. 11b,c). In Podzols, aluminum, iron, and organic compounds migrate from the surface soil with percolating rainwater down to deeper layers, resulting in a strongly leached surface soil[32]. The area in Southern Sweden that is the primary origin of accessions containing the FRO2-allele associated with slow growth under − Fe, shows a more diverse soil coverage (Supplementary Fig. 11b,d). There, the main soil classes are Cambisols in which there is no clear iron migration from upper layers and only minor iron leaching of surface soils[32]. Therefore, the northern Swedish accessions originate from an area with less iron abundance in the upper soil layers and the elevated FRO2 activity could be an adaptation to this iron limitation in the upper soil layers of Podzols.

In both, non-mutually exclusive, scenarios consistent with FRO2 allele distribution, the FRO2 alleles seem to facilitate a tuning of FRO2 expression that enables plants to grow even under limiting conditions. Being able to tune FRO2 expression is highly promising, as the simple overexpression of FRO2 has resulted in plant improvement even across species borders. Overexpression of FRO2 in transgenic soybean conferred improved Fe reduction in the roots and better tolerance towards IDC[33]. Overexpression of the yeast $Fe^{3+}$ chelate-reductase gene (FRE1) in rice led to an enhanced tolerance to low Fe availability in both hydroponic culture and calcareous soils and a yield of eight times more grain in calcareous soils[34]. The hope is that natural variants of FRO2 can be used for agronomic improvement of crop species by developing genotypes with longer root growth, better development, and subsequently enhanced tolerance toward IDC, especially with regard to different temperature ranges and in low Fe conditions, such as in podzols or calcareous soils.

## Methods

**Plant materials and growth conditions.** For surface sterilization, Arabidopsis thaliana seeds of 134 accessions from Sweden[21] that had been produced under uniform growth conditions were placed for 1 h in opened 1.5 ml Eppendorf tubes in a sealed box containing chlorine gas generated from 130 ml of 10% sodium hypochlorite and 3.5 ml of 37% hydrochloric acid. For stratification, seeds were imbibed in water and stratified in the dark at 4 °C for 3 days. Seeds were then put on the surface of 1 × MS agar plates, pH 5.7, containing 1% (w/v) sucrose and 0.8% (w/v) agar (Duchefa Biochemie) in 12 cm × 12 cm square plates (Greiner). Iron-deficient media (− Fe medium) is 1xMS medium (+ Fe medium) which instead of 100 μM Fe(III)-EDTA contains 300 μM FerroZine, a strong iron chelator [3-(2-pyridyl)-5,6-diphenyl-1,2,4-triazinesulfonate, Sigma-Aldrich][35–37]. To exclude confounding due to the presence of the chelator FerroZine, which is needed to chelate potential iron from the agar, we conducted the quantitative measurement of the mutants (Fig. 3b) on Fe-free medium that was prepared without FerroZine and with 1 × MS medium not containing Fe(III)-EDTA and with metal depleted agar. To obtain metal depleted agar, agar was washed three times with 50 mM EDTA, pH 5.7, for 8 to 16 h with continuous stirring, then washed six times with ultrapure water for at least 1 h and then used to prepare media[38]. The frd1-1 (fro2) mutant line is in the Col-0 background[22]. The T-DNA insertion line SAIL_411_E08 (T-DNA insertion in the first exon of the AT1G01570; Supplementary Fig. 7) was purchased from Nottingham Arabidopsis Stock Center (NASC, Nottingham, United Kingdom). The primers used for genotyping the T-DNA lines are shown in Supplementary Table 3a. Alkaline soil was prepared by addition of calcium oxide to a final soil pH of 7.0–8.0 (ref. 39). Plants were grown under a 16/8-h light-dark cycle at 21 °C.

**Root phenotyping and GWA mapping.** The 134 natural accessions (12 plants were used for each accession) were grown on − Fe agar plates under long day conditions (16 hours light) at 21 °C. Plant images were acquired with CCD flatbed scanners (EPSON Perfection V600 Photo, Seiko Epson CO., Nagano, Japan) every 24 hours for 5 days (2 DAG—6 DAG). Root image analyses and phenotype quantification were performed using the BRAT software[40]. Median total root

length values of 134 natural accessions were used for GWA studies. We conducted GWA mapping using a mixed model algorithm[41] which has been shown to correct for population structure confounding[42] and SNP data from the RegMap panel[28,43,44]. SNPs with minor allele counts greater or equal to 10 were taken into account. The significance of SNP associations was determined at 5% FDR threshold computed by the Benjamini-Hochberg-Yekutieli method to correct for multiple testing[45].

**Plasmid construction and plant transformation.** All plasmids were constructed using In-Fusion cloning technology (Clontech, Japan). The clonal vector was constructed using the pGreen0229 vector carrying a bacterial kanamycin selection and a plant basta (*nos-BAR*) resistance cassette as a backbone.

The pGreen0229 vector was modified by inserting an OCS terminator at the Xba I site. A *Pro35S:PM-cherry* reporter gene was amplified and inserted at the Xho I site. *Pro35S:PM-cherry* drives constitutive expression of a plasma-membrane localized mCherry, including in the mature seeds. The fluorescence of transformed seeds allowed us to select for positive transformants using a fluorescence microscope (Zeiss Discovery V8 or Leica MZ16FA stereomicroscope) without subjecting the plants to antibiotic selection[46].

For complementation, the *FRO2* gene, along with the *FRO2* native promoter (1,925 bp upstream of the translational start codon), was amplified from Boo2-3, T980, and TNY 04 (LR1 haplotype) from genomic DNA and cloned into the modified pGreen0229 vector at the EcoRV site. All plasmids were verified by sequencing and transformed into *fro2* mutant plants through Agrobacterium-mediated transformation[47]. The primers used for cloning are shown in Supplementary Table 3b.

**Quantitative real time PCR.** For qRT-PCR, plants were grown on 1xMS agar plates under normal growth conditions for 7 days after germination. Root tissues were collected by excision with fine scissors. Samples were immediately frozen in liquid nitrogen, ground, and total RNA was extracted using the RNeasy Plant Mini kit (QIAGEN GmbH, Hilden, Germany). qRT-PCR reactions were prepared using 2x SensiMix SYBR & Fluorescein Kit (PEQLAB LLC, Wilmington, DE, USA) and PCR was conducted with a Roche Lightcycler 96 (Roche) instrument. Relative quantifications were performed for all genes with the β-tubulin gene (AT5G62690) used as an internal reference. The primers used for qRT-PCR are shown in Supplementary Table 3c.

**Assay of ferric-chelate reductase activity.** Plants were grown for 2 weeks on $1 \times$ MS medium and then transferred to iron-sufficient or -deficient medium for 3 days. Ferric-chelate reductase activity was determined as described previously[15,23]. In brief, ten plants were pooled together as a group and whole roots of the plants were submerged in the assay solution containing 0.1 mM Fe(III)-EDTA and 0.3 mM FerroZine. After 30 min in the dark at room temperature, the roots were removed and the absorbance of assay solution was measured at 562 nm. Identical assay solution sample containing no plants was used as a blank. The concentration of Fe(II)-FerroZine was calculated using a molar extinction coefficient of $28.6 \, \text{mM}^{-1} \, \text{cm}^{-1}$.

**Haplotype analysis.** Haplotype analysis was performed using the previously described method[48]. Briefly, the 134 natural accessions of *A. thaliana* were grouped according to haplotypes based on the SNPs in a 50 kb window around the *FRO2* locus. SNPs in a 50 kb window around the *FRO2* locus were extracted from a pre-imputation version of the Regional Mapping Project SNP panel described in Horton et al.[28] These SNPs were used as the input for fastPHASE version 1.4.0 (ref. 49) and results were analysed and visualized using custom R scripts.

**Climate and soil data.** With regard to climate variable analyses, we mainly focused on a subset of the climate data gathered by Hancock et al.[50]. Raw data of climate variables (19 climate, latitude and longitude) were downloaded from the WorldClim project (www.worldclim.org)[51]. A linear model was used to test correlations between each of the variables and the root length phenotype. *P*-values were corrected for multiple testing. We obtained weak but significant correlations with temperature (Supplementary Table 1). However, this correlation was lost when taking into account population structure (Supplementary Table 2). To correct for population structure, we added to the linear model the first two eigenvectors that were obtained by performing a PCA on the SNP matrix that was used to run GWAS. PCA analyses were conducted with the eigenstrat() function from the AssocTests package in the R programming environment[52].

For soil-type analysis we used GIS-maps from the European soil database[53,54]. These maps were visualized and analysed in QGIS (http://qgis.osgeo.org)[55].

**LFMM analysis.** We performed a latent factor mixed model (LFMM) analysis to find significant associations with temperature variable while accounting for population structure[27]. We used the Bio01 (mean annual temperature)[56] of the point of origin of all accessions of the RegMap panel[28] as an input and the SNP matrix that we had used for GWAS as genotype matrix. We then ran LFMM[27]

using the default setting and calculated the median Z-score from the single Z-Scores of four independent runs.

**Data availability.** The authors declare that all data supporting the findings of this study can be found within the paper and its Supplementary Files. Additional data supporting the findings of this study are available from the corresponding author (W.B.) upon request.

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

## Acknowledgements

We thank Mary Lou Guerinot (Dartmouth College, USA) and Erin Connolly (University of South Carolina, USA) for providing *FRO2* null mutant (*frd1-1*) and *FRO2* over-expression line seeds. We are grateful to M. Nordborg (GMI, Vienna) for donating seeds of natural accessions and early access to SNP and genome sequences related data, Ümit Seren (GMI, Vienna) for help with providing SNPs and Fernando Rabanal (GMI, Vienna) for providing de novo assembly data of TNY 04. We are also grateful to A. Hancock for discussions on the relation of SNPs and environmental variables and to C. Goeschl for installing and running LFMM. We are grateful to members of Busch laboratory for critical reading of the manuscript and M. Watson for manuscript editing. We thank B. Wohlrab, K. Jandrasits, S. Krasnodebski, S. Perez, D. Stoeva, and F. Stritzinger for technical assistance. This work was supported by funds from the Austrian Academy of Science through the Gregor Mendel Institute (GMI) and an Austrian Science Fund (FWF) stand-alone project (P27163-B22).

## Author contributions

S.B.S. performed most experiments. C.S. helped conduct RNA extraction, expression analysis and Sanger sequencing. F.F. helped conducting the −Fe root phenotyping screen. R.S. performed +Fe root phenotyping. E.K. conducted haplotype and environmental variable correlation analyses. S.B.S. and W.B. designed the research, analysed data and wrote the manuscript. W.B. supervised this project.

## Additional information

**Competing interests:** The authors declare no competing financial interests.

