## [Peer Review File · Nature Communications]

Reviewers' comments:

Reviewer #1 (Remarks to the Author):

This manuscript describes experiments designed to identify genes/alleles responsible for natural variation in root growth responses to iron. The authors used a set of *Arabidopsis thaliana* accessions from Sweden from contrasting environments to detect natural variation in root length under Fe deficiency. GWAS allowed them to identify FRO2 as a major determinant of root length under -Fe conditions. In addition, the authors show that the distribution of FRO2 alleles is associated with temperature.

The work reported by Satbhai et al is novel and significant. While the role of FRO2 in the reduction of rhizosphere Fe for subsequent uptake by IRT1 has been known for some time, a role for FRO2 in control of root growth is new. In general, the manuscript is clear and easy to follow and the data are (for the most part) convincing. However, I have several concerns about the manuscript as it stands now. One concern I have is that FRO2 resides in tandem on the chromosome with another Fe deficiency regulated gene (FRO1- At1g01590). To rule out a role for FRO1 in the control of root growth under -Fe, the authors should (at the minimum) measure FRO1 expression levels across accessions with short or long roots under -Fe.

With regard to complementation of *frd1* with FRO2 genomic constructs: The authors do not provide any information on how much promoter was included in the constructs. Are the promoter region SNPs for the LR1 and LR2 haplotypes included in the complementation constructs? The authors should follow-up with complementation constructs that have individual polymorphisms to identify the causal changes. The authors also should measure average FRO2 expression levels for each set of transgenic lines to confirm that the phenotype is due to FRO2 expression level (as suggested by Fig. 1E). Another missing piece is the link to FRO2 activity. The authors should determine whether or not the phenotype is correlated with ferric chelate reductase activity.

Specific comments:

1. Line 52: phytosiderophores chelate Fe³⁺.
2. Line 61: FRO2 stands for Ferric Reductase Oxidase.
3. Line 62: FRO2 reduces ferric iron- chelates to Fe²⁺.
4. After the mention of AHA2 and FRO2, the authors should mention IRT1.
5. Line 100: add a period (.) after Supplementary Figure 2.
6. Line 111-112: The authors refer to Supplementary Table 3 here, but I'm not sure it's relevant here.
7. Line 131: *fro2* mutants were originally named *frd1*.
8. Line 141: add "the" before FRO2 gene.
9. Figure 2C: The phenotype shown is not quite consistent with what one expects for plants grown on alkaline soil. The plants are smaller, but not necessarily more chlorotic than when grown under pH 7.0. In addition, the quality of the photo is not sufficient for publication.

Reviewer #2 (Remarks to the Author):

Summary: The authors assessed root growth under iron deficiency for 134 Swedish accessions of *Arabidopsis thaliana* and found significant natural variation. A GWAS analysis identified FRO2 as the best candidate for the observed variation in root growth response. This is certainly interesting, as FRO2 had previously been shown to catalyze the rate-limiting step in iron uptake, the reduction of Fe³⁺ to Fe²⁺. The authors go on to suggest that FRO2, in addition to its biochemical function, also has a regulatory role to play but what this role might be remains a mystery. They also suggest a possible adaptive role for FRO2 in response to temperature based on a "peculiar" SNP pattern seen in the RegMap panel. I find that several key experiments are missing that are needed to validate the conclusions being drawn.

1. The authors report that FRO2 expression levels correlate with root length (Figure 1) but they have not reported on the ferric chelate reductase activity of their accessions. They need to determine whether changes in gene expression result in changes in levels of the FRO2 protein and/or to changes in ferric chelate reductase activity. This is especially important as the authors are suggesting a regulatory role for FRO2. The ferric chelate reductase activity assay is simple and fast and can be carried out with whole plants.

2. Complementation (Figure 3): For their complemented lines, the authors only report root length. As stated above, they need to measure FRO2 expression and ferric chelate reductase activity.

3. The authors identified two haplotypes among the accessions with the longest roots, each with two non-coding SNPs. Why is this information presented in supplementary figures (Supplementary figures 7 and 8) as it seems central to their argument that they have identified the causal SNPs. Of the accessions used for creating the complementation clones (Boo2-3, TNY-04 and T980) only TNY-04 is included in Supplementary Figure 7, presumably because the other two accessions were not available in the SALK 1001 genomes browser. I think the authors should add the results of their sequencing of the Boo2-3 and T980 accessions to Supplementary figure 7. It is almost impossible to see the causal SNPs in Supplementary figure 8 and the reader shouldn't have to be flipping back and forth to see which haplotype pattern they have. The authors speculate in their discussion that the SNPs might affect FRO2 transcription and/or affect FRO2 RNA post-transcriptionally. Why speculate? This can easily be determined.

4. The authors argue that because the absence of Fe does not greatly restrict the growth of WT roots at the particular time point examined whereas *fro2* plants have much shorter roots under these same conditions that FRO2 has a regulatory role in addition to its biochemical role. Am I missing something? WT is able to take up enough Fe because FRO2 is functional and a *fro2* plant quickly becomes limited for Fe as reduction is the rate-limiting step for Fe uptake. One does not need to invoke a regulatory role for FRO2 to explain these results. If FRO2 really has a non-enzymatic role, then expressing an enzymatically inactive form of

FRO2 should presumably "rescue" the root length phenotype.

5. The co-variation of mean temperature values and median root length on -Fe on day 2 is an interesting observation but it seems premature to be presenting it here. Supplementary Figure 9 is shown in support of their observation that long root accessions were more commonly associated with colder temperatures than accessions with shorter roots but, as the authors point out, the correlation is non-significant once population structure and correction for multiple testing are taken into account. Thus, there is no correlation.

6. What exactly do the authors mean when they call the SNP pattern "peculiar" (p.10, line 240)? Is the difference shown in Supplement Figure 10 statistically significant?

Reviewer #3 (Remarks to the Author):

The authors of this paper conduct a traditional GWAS for a single interesting trait within *A. thaliana*: root length under low iron conditions. The results unambiguously point to a locus adjacent to a known candidate gene which is confirmed with several follow-up complementation analyses. The conclusion that FRO2, a gene named for its involvement in iron uptake, naturally harbors multiple regulatory variants that confer varying root-growth responses in iron-deficient environments, is quite strong.

Further analyses of the geographic distribution of these alleles is under-powered to demonstrate the action of local adaptation, and the authors' conclusion that environmental correlates provide "a hint for an adaptive role" of this variation is not compelling.

This paper could benefit from additional analysis: the construction of a plant with an inducible FRO2 gene could confirm that FRO2 expression levels completely explain the phenotype, and better yet would a demonstration that FRO2 expression generates an iron-dependent fitness difference. A proper population genetic analysis could look for population-specific signatures of selection on the identified alleles, and better yet could yield testable predictions of the environment-specific fitness differentials of these alleles.

The paper is clearly-written but could be improved by review for idiomatic English usage.

We would like to thank the reviewers for their constructive feedback and their excellent suggestions that helped to improve the manuscript. We were able to address the issues raised by the reviewers by additional experiments as well as editing the manuscript. Main changes involve additional experiments that test ferric chelate reductase activity in the accessions and the complemented lines, as well as the determination of *FRO2* expression level in the lines used for allelic complementation. These new experiments allowed us to make our conclusions regarding the effect of the alleles on the expression level of *FRO2* and the ferric chelate reductase activity much stronger. Moreover, we have removed emphasis from the adaptive value of *FRO2* alleles to avoid misconceptions. While not requested by the reviewers, we converted all presented root length data to mm **instead of pixels to spare interested readers a unit conversion**. Here we provide a point-by-point response to the issues raised by the reviewers.

Reviewer #1

- This manuscript describes experiments designed to identify genes/alleles responsible for natural variation in root growth responses to iron. The authors used a set of Arabidopsis thaliana accessions from Sweden from contrasting environments to detect natural variation in root length under Fe deficiency. GWAS allowed them to identify FRO2 as a major determinant of root length under -Fe conditions. In addition, the authors show that the distribution of FRO2 alleles is associated with temperature.

The work reported by Satbhai et al is novel and significant. While the role of FRO2 in the reduction of rhizosphere Fe for subsequent uptake by IRT1 has been known for some time, a role for FRO2 in control of root growth is new. In general, the manuscript is clear and easy to follow and the data are (for the most part) convincing. However, I have several concerns about the manuscript as it stands now.

One concern I have is that FRO2 resides in tandem on the chromosome with another Fe deficiency regulated gene (FRO1- At1g01590). To rule out a role for FRO1 in the control of root growth under -Fe, the authors should (at the minimum) measure FRO1 expression levels across accessions with short or long roots under -Fe.

We have now ruled out a role of FRO1 by measuring its expression level in the groups of accessions as proposed by the reviewer (Supplementary Figure 4, Line 139-148).

With regard to complementation of frd1 with FRO2 genomic constructs: The authors do not provide any information on how much promoter was included in the constructs. Are the promoter region SNPs for the LR1 and LR2 haplotypes included in the complementation constructs?

We have included the requested information in the materials and methods section (Line 403-405).

The authors should follow-up with complementation constructs that have individual polymorphisms to identify the causal changes.

We agree with the reviewer that this would be a very interesting experiment. We believe however, that this is outside the scope of this manuscript as we have clearly shown that the observed variation is due to the different alleles. Getting to the individual SNPs would only reconfirm this conclusion but add many months of additional experimentation.

The authors also should measure average FRO2 expression levels for each set of transgenic lines to confirm that the phenotype is due to FRO2 expression level (as suggested by Fig. 1E).

We have now measured the FRO2 expression level in our set of transgenic lines (Fig. 6c). This allowed us to significantly strengthen our conclusions (page 11).

Another missing piece is the link to FRO2 activity. The authors should determine whether or not the phenotype is correlated with ferric chelate reductase activity.

We now have measured the Ferric Chelate Reductase activity in extreme accessions under +Fe and -Fe conditions (**Fig. 2a**) and found that elevated ferric chelate reductase activity is highly correlated with total root length under -Fe conditions (**Fig. 2b**) as well as *FRO2* transcript levels (**Fig. 2c**). Additionally, we have also conducted these assays in our set of transgenic lines (**Fig. 6d**) and have also grown these lines on alkaline pH soil to check the effect of allelic variation on growth (**Fig. 6e**).

Specific comments:

1. Line 52: phytosiderophores chelate Fe³⁺.

Corrected (**Line 55**).

2. Line 61: FRO2 stands for Ferric Reductase Oxidase.

Corrected (**Line 65**).

3. Line 62: FRO2 reduces ferric iron- chelates to Fe²⁺.

Corrected (**Line 67**).

4. After the mention of AHA2 and FRO2, the authors should mention IRT1.

Corrected (**Line 67**).

5. Line 100: add a period (.) after Supplementary Figure 2.

Corrected (**Line 104**).

6. Line 111-112: The authors refer to Supplementary Table 3 here, but I'm not sure it's relevant here.

Corrected.

7. Line 131: *fro2* mutants were originally named *frd1*.

Corrected (**Line 137**).

8. Line 141: add "the" before FRO2 gene.

Corrected (**Line 167**).

9. Figure 2C: The phenotype shown is not quite consistent with what one expects for plants grown on alkaline soil. The plants are smaller, but not necessarily more chlorotic than when grown under pH 7.0. In addition, the quality of the photo is not sufficient for publication.

We have conducted the growth experiment again and improved the quality of the picture by using a better camera (**Fig. 3c**). Chlorotic leaves can now be easily spotted in many plants of the *fro2* mutant line when there are still no chlorotic leaves in the WT.

Reviewer #2 (Remarks to the Author):

Summary: The authors assessed root growth under iron deficiency for 134 Swedish accessions of Arabidopsis thaliana and found significant natural variation. A GWAS analysis identified FRO2 as the best candidate for the observed variation in root growth response. This is certainly interesting, as FRO2 had previously been shown to catalyze the rate-limiting step in iron uptake, the reduction of Fe³⁺ to Fe²⁺. The authors go on to suggest that FRO2, in addition to its

biochemical function, also has a regulatory role to play but what this role might be remains a mystery. They also suggest a possible adaptive role for FRO2 in response to temperature based on a "peculiar" SNP pattern seen in the RegMap panel. I find that several key experiments are missing that are needed to validate the conclusions being drawn.

1. The authors report that FRO2 expression levels correlate with root length (Figure 1) but they have not reported on the ferric chelate reductase activity of their accessions. They need to determine whether changes in gene expression result in changes in levels of the FRO2 protein and/or to changes in ferric chelate reductase activity. This especially important as the authors are suggesting a regulatory role for FRO2. The ferric chelate reductase activity assay is simple and fast and can be carried out with whole plants.

We now have measured the ferric chelate reductase activity in extreme accessions under +Fe and -Fe conditions (**Fig. 2a**) and found that elevated ferric chelate reductase activity is highly correlated with the total root length under -Fe conditions (**Fig. 2b**) as well as FRO2 transcript levels (**Fig. 2c**). Additionally, we have also conducted these assays in our set of transgenic lines (**Fig. 6d**) and have also grown these lines on alkaline pH soil to check the effect of allelic variation on growth (**Fig. 6e**).

2. Complementation (Figure 3): For their complemented lines, the authors only report root length. As stated above, they need to measure FRO2 expression and ferric chelate reductase activity.

We have now measured the FRO2 expression level in our set of transgenic lines (**Fig. 6**). This allowed us to significantly strengthen our conclusions. As mentioned above, we also measured ferric chelate reductase in these lines.

3. The authors identified two haplotypes among the accessions with the longest roots, each with two non-coding SNPs. Why is this information presented in supplementary figures (Supplementary figures 7 and 8) as it seems central to their argument that they have identified the causal SNPs. Of the accessions used for creating the complementation clones (Boo2-3, TNY-04 and T980) only TNY-04 is included in Supplementary Figure 7, presumably because the other two accessions were not available in the SALK 1001 genomes browser. I think the authors should add the results of their sequencing of the Boo2-3 and T980 accessions to Supplementary figure 7. It is almost impossible to see the causal SNPs in Supplementary figure 8 and the reader shouldn't have to be flipping back and forth to see which haplotype pattern they have.

We have moved the previous Supplementary Figure 7 to main **Figure 5**. We have also displayed the polymorphisms in the Boo2-3, T980, and TNY 04 accessions based on Sanger sequencing as an easily readable figure (Supplementary Fig. 9).

The authors speculate in their discussion that the SNPs might affect FRO2 transcription and/or affect FRO2 RNA post-transcriptionally. Why speculate? This can easily be determined.

As mentioned above, we have now measured the FRO2 transcript levels in all lines.

4. The authors argue that because the absence of Fe does not greatly restrict the growth of WT roots at the particular time point examined whereas fro2 plants have much shorter roots under these same conditions that FRO2 has a regulatory role in addition to its biochemical role. Am I missing something? WT is able to take up enough Fe because FRO2 is functional and a fro2 plant quickly becomes limited for Fe as reduction is the rate-limiting step for Fe uptake. One does not need to invoke a regulatory role for FRO2 to explain these results. If FRO2 really has a non-enzymatic role, then expressing an enzymatically inactive form of FRO2 should presumably "rescue" the root length phenotype.

We thank the reviewer for pointing the lack of clarity of our argument. We have now included

following addition to the results to make this clearer (Line 191-195).

“... We note, that it is highly unlikely to be due to the enzymatic activity of FRO2, as there is no iron available in the medium for FRO2 to reduce. Yet, under these -Fe conditions, 35S:FRO2 plants display longer roots than WT and fro2 mutant plants display shorter roots than WT. Therefore, this suggests a regulatory role of FRO2 in addition to its known enzymatic role.”

We also rephrased and extended this point in the discussion (Line 291-295).

“... As there is no iron for WT or fro2 mutants to take up, we would expect similar growth effects in both lines if the phenotype was related to iron availability. As the phenotypes were remarkably different, this therefore suggests that FRO2 has a regulatory function in addition to its biochemical function, as the absence of Fe alone does not dramatically restrict root growth at this stage of root development..”

5. The co-variation of mean temperature values and median root length on -Fe on day 2 is an interesting observation but it seems premature to be presenting it here. Supplementary Figure 9 is shown in support of their observation that long root accessions were more commonly associated with colder temperatures than accessions with shorter roots but, as the authors point out, the correlation is non-significant once population structure and correction for multiple testing are taken into account. Thus, there is no correlation.

6. What exactly do the authors mean when they call the SNP pattern "peculiar" (p.10, line 240)? Is the difference shown in Supplement Figure 10 statistically significant?

The reviewer raised good points: While there is a correlation between climate and -Fe dependent root growth in the Swedish population, it is not significant when taking population structure into account. However, one can expect that the strong population structure in Sweden leads to a high number of false negatives. This is why we looked at the worldwide population. Because we don't have sufficient root growth data for the worldwide population under -Fe, we looked at SNP patterns. However, as pointed out by this reviewer, this is a bit fuzzy. While just doing a statistical test leads to a very significant correlation, this could as well be due to population structure. We therefore performed an analysis that correlated SNPs to climate variables for all lines of the RegMap panel and that also accounted for population structure. When tested only for the FRO2 marker SNP we observed a significant correlation of temperature and the FRO2 marker SNP from the GWAS, however this was not significant at the whole genome level. Based on these analyses, we deemphasized the potential adaptive value and moved the whole section and the new analysis to the discussion, where, together with the known role of FRO2 in chilling tolerance, it is still a very interesting aspect (Line 327-335).

Reviewer #3 (Remarks to the Author):

The authors of this paper conduct a traditional GWAS for a single interesting trait within A. thaliana: root length under low iron conditions. The results unambiguously point to a locus adjacent to a known candidate gene which is confirmed with several follow-up complementation analyses. The conclusion that FRO2, a gene named for its involvement in iron uptake, naturally harbors multiple regulatory variants that confer varying root-growth responses in iron-deficient environments, is quite strong.

Further analyses of the geographic distribution of these alleles is under-powered to demonstrate the action of local adaptation, and the authors' conclusion that environmental correlates provide "a hint for an adaptive role" of this variation is not compelling.

We conducted additional analyses and concluded that no strong point can be made with the data that we have. Based on these analyses, we deemphasized the potential adaptive value and moved the whole section and the new analysis to the discussion, where, together with the known role of FRO2 in chilling tolerance, it is still a very interesting aspect (Line 316-335).

This paper could benefit from additional analysis: the construction of a plant with an inducible FRO2 gene could confirm that FRO2 expression levels completely explain the phenotype, and better yet would a demonstration that FRO2 expression generates an iron-dependent fitness difference.

We have now conducted additional experiments using a *FRO2* overexpressing line and shown that *FRO2* expression levels explain the root phenotype (**Supplementary Fig. 5, line 186-190**).

A proper population genetic analysis could look for population-specific signatures of selection on the identified alleles, and better yet could yield testable predictions of the environment-specific fitness differentials of these alleles.

As we have deemphasized the adaption part now, we reasoned such an ambitious analysis would go beyond the scope of the manuscript and leave it for follow up studies to explore this.

The paper is clearly-written but could be improved by review for idiomatic English usage.

Our institute's scientific editor who is a native English speaker has now edited the manuscript once more.

Reviewers' comments:

Reviewer #1 (Remarks to the Author):

The authors have done a nice job of responding to my comments on the first version of the manuscript. They have addressed my concerns regarding FRO1 and have included new data (FRO2 transcript levels and ferric chelate reductase activity) that significantly strengthen the major conclusion of the manuscript. While I feel that it is important to identify the causal SNPs in order to get a better idea for the underlying mechanism, I agree that this manuscript represents a significant advance on its own. Indeed, the authors have shown that FRO2 is a clear determinant of root length under -Fe which constitutes an important insight that should be of considerable interest in the field.

Reviewer #4 (Remarks to the Author):

In their study Satbhai et al have identified (GWAS analysis) FRO2 as the causal gene associated with variation in root length of Swedish Arabidopsis accessions growing on a medium deprived of iron (Fe). Based on their experiments the authors propose that FRO2 has a regulatory activity that is independent of its enzymatic activity. Last, the authors have identified various non-coding SNPs that influence FRO2 mRNA steady states (the molecular mechanisms involved remaining to be identified and characterized).

If the role of FRO2 in influencing the root length of the Swedish Arabidopsis population when grown under Fe deficiency is new and to some extent interesting, the findings presented in the manuscript do not deeply change what is already known on the role of FRO2 and the Arabidopsis Fe acquisition machinery.

One of the most original points of the Satbhai et al manuscript is the claim that FRO2 play a regulatory role that is independent of its enzymatic activity in the plant response to Fe deprivation. Unfortunately, the data that are presented do not support this claim and rather support the already known function of FRO2 in the IRT1/FRO2 Fe acquisition machinery.

- First it would be necessary to demonstrate that an inactive form of FRO2 (i.e. deprived of its enzymatic activity) still plays a role in the plant response to Fe deprivation. This is for example what has been done to demonstrate the regulatory role of the hexokinase HXK1 where the authors have shown that "...mutants lacking catalytic activity still support various signaling functions in gene expression, cell proliferation, root and inflorescence growth..." (see Moore et al 2003, Science). In the case of the Satbhai et al work it could be for example to show that the expression of a FRO2 gene lacking its reductase activity is able to rescue the fro2 phenotype (or at least part of it) when grown in Fe deficiency.

- Second, the authors should show that the decrease in primary root growth observed when the fro2 mutant is grown under Fe deficiency is specific when compared to the other genes associated with the Arabidopsis Fe acquisition machinery. In this regard IRT1 is the most obvious gene as its expression is controlled by the same set of regulatory genes as FRO2, and thus both genes display the same expression pattern (see for e.g. see Zhang et al 2015, Plant Cell. & Li et al 2016, Plant Physiol & Brumbarova et al 2015, TIPS). One

would have included the *irt1* mutant in the analysis as a control genotype. However, various studies have already shown that *irt1* mutant display, like *fro2* in the Satbhai et al work, a shorter roots when grown in Fe deficiency (see for e.g. Zhang et al 2014, New Phytol). This is also the case of FIT, a direct transcriptional regulator of IRT1 and FRO2 expression. Similarly, the loss-of-function mutants of bHLH34, bHLH104 and bHLH105 transcription factors that act upstream FIT display a similar root phenotype when grown under Fe deficiency. Altogether these data indicate that FRO2 do not possess a specific regulatory function. In fact the reduction of the primary root length of the *fro2* mutant is not “new” even if it has been poorly directly reported mostly because people in the field rather use IRT1 as the model gene for the Fe acquisition mechanism.

- Another important point is that within the first few days following germination, seedlings rely on the nutrients that have been stored during the seed filling while they were on the mother plants. In this case, *fro2* mother plants growing on the same media as wild type plants may accumulate less Fe in their seeds that will be detrimental to the embryo upon germination. This may impact the growth of the seedlings after germination, before the seedlings starts to mine its substrate in search of nutrients. However, in order to improve the seed yield of the *fro2* or the *irt1* mutants, it is frequent to supplement the medium with exogenous Fe (usually a form that is easily assimilated by the plant, such Fe-EDDHA). Thus the mother plant and their seeds can get more Fe. Nevertheless, the cell wall/apoplasmic space is a compartment of the cell that contains large proportion of Fe (in addition to the vacuole) from which the IRT1/FRO2 (AHA2/PDR9) machinery can get substantial amount of Fe (see for e.g., Curie and Mari, 2016 - New Phytol). In the case of the *fro2* mutant this machinery is indeed not operating. In other words overexpressing a functional FRO2 in *fro2* will allow the seedling to access to this Fe before mining the substrate, whereas overexpressing a functional FRO2 in wild type won't have much effect.

The Col-0 FRO2 activity is lacking Figure 2 because it is known that the Col-0 FRO2 activity is strongly induced in response to Fe deficiency (see for e.g., Robinson et al 1999, Nature) like what is observed for Group2 alleles (long roots) in Figure 2. This is in contradiction with the results presented Figure 1 where the expression of FRO2 in Col-0 is similar to that of Group1 alleles (short roots). This point should be discussed and Col-0 FRO2 activity included Figure 2. Another intriguing observation that is not discussed is that Figure 2 the FRO2 activity is not significantly different between the +Fe and -Fe conditions for the Group1 alleles (including Boo2-3) whereas there is a clear induction in Figure 6d for the Boo2-3 allele.

Within the discussion it is proposed that FRO2 mRNA steady state in the studied population may be associated with (low) temperature, but is there any information on soil Fe amount, availability or soil pH in the area where these accessions have been collected. In addition, it would have been interesting to clearly stated what was the rationale of choosing this population to study the impact of micronutrients availability (Fe deprivation) on Arabidopsis root growth.

We would like to thank the reviewers for their effort to evaluate and improve our manuscript. We addressed all important issues that were raised by the reviewers by additional analyses as well as by editing the manuscript.

Here we provide a description of the changes by a point-by-point response to the issues raised by the reviewers.

Reviewer #1 (Remarks to the Author):

The authors have done a nice job of responding to my comments on the first version of the manuscript. They have addressed my concerns regarding FRO1 and have included new data (FRO2 transcript levels and ferric chelate reductase activity) that significantly strengthen the major conclusion of the manuscript. While I feel that it is important to identify the causal SNPs in order to get a better idea for the underlying mechanism, I agree that this manuscript represents a significant advance on its own. Indeed, the authors have shown that FRO2 is a clear determinant of root length under -Fe which constitutes an important insight that should be of considerable interest in the field.

Response to Reviewer #1:

We would like to thank the reviewer #1 for his/her review. We are happy that the reviewer agrees with us that our manuscript represents a significant advance on its own, even when not mechanistically studying the causal SNPs (something that we think would be out of the scope of this paper).

Reviewer #4 (Remarks to the Author):

In their study Satbhai et al have identified (GWAS analysis) FRO2 as the causal gene associated with variation in root length of Swedish Arabidopsis accessions growing on a medium deprived of iron (Fe). Based on their experiments the authors propose that FRO2 has a regulatory activity that is independent of its enzymatic activity. Last, the authors have identified various non-coding SNPs that influence FRO2 mRNA steady states (the molecular mechanisms involved remaining to be identified and characterized).

If the role of FRO2 in influencing the root length of the Swedish Arabidopsis population when grown under Fe deficiency is new and to some extent interesting, the findings presented in the manuscript do not deeply change what is already known on the role of FRO2 and the Arabidopsis Fe acquisition machinery.

One of the most original points of the Satbhai et al manuscript is the claim that FRO2 play a regulatory role that is independent of its enzymatic activity in the plant response to Fe deprivation. Unfortunately, the data that are presented do not support this claim and rather

support the already known function of FRO2 in the IRT1/FRO2 Fe acquisition machinery.

- First it would be necessary to demonstrate that an inactive form of FRO2 (i.e. deprived of its enzymatic activity) still plays a role in the plant response to Fe deprivation. This is for example what has been done to demonstrate the regulatory role of the hexokinase HXK1 where the authors have shown that "...mutants lacking catalytic activity still support various signaling functions in gene expression, cell proliferation, root and inflorescence growth..." (see Moore et al 2003, Science). In the case of the Satbhai et al work it could be for example to show that the expression of a FRO2 gene lacking its reductase activity is able to rescue the fro2 phenotype (or at least part of it) when grown in Fe deficiency.

*We thank the reviewer for this good suggestion. We agree with the reviewer that this would be a very interesting experiment. We believe however, that this is outside the scope of this manuscript. Moreover, because of the other important issues that the reviewer raised (see below), we decided to tone down the conclusion about the additional regulatory role of the FRO2 protein and provide potential alternative explanation that takes into account seed loading with iron (as suggested by the reviewer, **Line 305-314**).*

- Second, the authors should show that the decrease in primary root growth observed when the fro2 mutant is grown under Fe deficiency is specific when compared to the other genes associated with the Arabidopsis Fe acquisition machinery. In this regard IRT1 is the most obvious gene as its expression is controlled by the same set of regulatory genes as FRO2, and thus both genes display the same expression pattern (see for e.g. see Zhang et al 2015, Plant Cell. & Li et al 2016, Plant Physiol & Brumbarova et al 2015, TIPS). One would have included the irt1 mutant in the analysis as a control genotype. However, various studies have already shown that irt1 mutant display, like fro2 in the Satbhai et al work, a shorter roots when grown in Fe deficiency (see for e.g. Zhang et al 2014, New Phytol). This is also the case of FIT, a direct transcriptional regulator of IRT1 and FRO2 expression. Similarly, the loss-of-function mutants of bHLH34, bHLH104 and bHLH105 transcription factors that act upstream FIT display a similar root phenotype when grown under Fe deficiency. Altogether these data indicate that FRO2 do not possess a specific regulatory function. In fact the reduction of the primary root length of the fro2 mutant is not "new" even if it has been poorly directly reported mostly because people in the field rather use IRT1 as the model gene for the Fe acquisition mechanism.

*The reviewer raised very good points. Indeed, mutants for IRT1, FIT1 and the mentioned BHLH factors display shorter roots under Fe deficiency condition. We agree that this provides support for an alternative model explaining the shorter roots due to Fe loading into the seeds (see below). We have incorporated this now in the manuscript (**Line 305-314**). We'd like to point out that none of these studies had quantified the root lengths. More importantly, there is no report related to a short root phenotype of fro2 mutants under -Fe, which is an important point of this study, especially with regard to the key finding of this manuscript, that natural allelic variation of FRO2 is a major determinant of Arabidopsis root growth under -Fe in natural accessions. Overall, we think the occurrence of the short root length phenotypes under -Fe for some other Fe related genes, does not impact our central finding that natural allelic variation of FRO2 that leads to*

FRO2 expression and activity differences causes altered root growth under iron limited conditions.

- Another important point is that within the first few days following germination, seedlings rely on the nutrients that have been stored during the seed filing while they were on the mother plants. In this case, *fro2* mother plants growing on the same media as wild type plants may accumulate less Fe in their seeds that will be detrimental to the embryo upon germination. This may impact the growth of the seedlings after germination, before the seedlings starts to mine its substrate in search of nutrients. However, in order to improve the seed yield of the *fro2* or the *irt1* mutants, it is frequent to supplement the medium with exogenous Fe (usually a form that is easily assimilated by the plant, such Fe-EDDHA). Thus the mother plant and their seeds can get more Fe. Nevertheless, the cell wall/apoplasmic space is a compartment of the cell that contains large proportion of Fe (in addition to the vacuole) from which the IRT1/FRO2 (AHA2/PDR9) machinery can get substantial amount of Fe (see for e.g., Curie and Mari, 2016 - New Phytol). In the case of the *fro2* mutant this machinery is indeed not operating. In other words overexpressing a functional FRO2 in *fro2* will allow the seedling to access to this Fe before mining the substrate, whereas overexpressing a functional FRO2 in wild type won't have much effect.

The reviewer raised a very interesting point and we do agree that seed nutrient content is an important factor for early growth following germination. We have now added this additional model that could explain the fro2 short-root phenotype under -Fe to the manuscript (Line 305-314).

The Col-0 FRO2 activity is lacking Figure 2 because it is known that the Col-0 FRO2 activity is strongly induced in response to Fe deficiency (see for e.g., Robinson et al 1999, Nature) like what is observed for Group2 alleles (long roots) in Figure 2. This is in contradiction with the results presented Figure 1 where the expression of FRO2 in Col-0 is similar to that of Group1 alleles (short roots). This point should be discussed and Col-0 FRO2 activity included Figure 2. Another intriguing observation that is not discussed is that Figure 2 the FRO2 activity is not significantly different between the +Fe and -Fe conditions for the Group1 alleles (including Boo2-3) whereas there is a clear induction in Figure 6d for the Boo2-3 allele.

We thank the reviewer for pointing the lack of clarity in Figure 2a related to the FRO2 activity measurements under +Fe and -Fe conditions using extreme accessions. This stemmed from our poor way of averaging the data for that figure. We had compared all the values (not just the induction) between the two groups without regard for the individual accession. This lead to a clear underestimation of the effect and we are thankful that the reviewer picked this up. We have now depicted values for the individual accessions and indicated the averages induction (fold change) upon -Fe. From this comparison, it is clearly visible now that FRO2 activity is induced in both groups, but that the induction in group 1 is significantly lower than in group 2 (Figure 2).

We did not include Col-0 accession in this experiment as we assessed the activity from set of

extreme accessions originated in Sweden, which were used for phenotyping screen. Perhaps, this would have been crucial if there would be no induction in one group (as the reviewer had assumed), but as this is not the case we don't need Col-0 as a positive control.

Within the discussion it is proposed that FRO2 mRNA steady state in the studied population may be associated with (low) temperature, but is there any information on soil Fe amount, availability or soil pH in the area where these accessions have been collected. In addition, it would have been interesting to clearly stated what was the rationale of choosing this population to study the impact of micronutrients availability (Fe deprivation) on Arabidopsis root growth.

We have added a rationale to choosing this population in the beginning of the results section (Supplementary Figure 1, Line 98-104).

Encouraged by the reviewer, we have now conducted an analysis on soil type with the occurrence of the FRO2 alleles. Interestingly, we found that the area from which most fast growing G-accessions come from is covered in Podzol in which aluminium, iron and organic compounds migrate from the surface soil down to deeper layers with percolating rainwater (Supplementary Figure 12b). In contrast, Southern Sweden area that mostly contains the slower growing A-accessions shows a more diverse soil structure. Another potentially adaptive role could be attributed to the north/south difference in soil types (Supplementary Figure 12c; Line 353-366).

REVIEWERS' COMMENTS:

Reviewer #4 (Remarks to the Author):

In their study Satbhai et al have identified (GWAS analysis) FRO2 as the causal gene associated with variation in root length of Swedish Arabidopsis accessions growing on a medium deprived of iron (Fe).

Even if the work has been very nicely done and the results are of some interest for the plant community, the findings that are presented do not change what is already known on the role of FRO2 and the Arabidopsis Fe acquisition machinery/Fe homeostasis. This is rather confirming what is already known.

Because GWAS studies are now widely used to identify the genes regulating the studied traits (e.g. Angelovici et al 2016 - Plant Physiol; Kloth et al 2016 - J Exp Bot; Julkowska et al 2016 - J Exp Bot) and because FRO2 is known for long to be involved in plant Fe homeostasis and related development (Robinson et al 1999 - Nature), the extent of the novelty that is presented in this manuscript is rather low.

What would have made this manuscript outstanding would have been either (i) to decipher why non-coding SNPs influence FRO2 mRNA steady states or (ii) to demonstrate that FRO2 acts as a regulatory protein. Unfortunately none of these two options has been successfully pursued (i.e. no novel data were included in the revised document) and thus limits the importance of the study.